# Wind-farm layout optimisation using a hybrid Jensen–LES approach

Vahid S. Bokharaie, Pieter Bauweraerts, and Johan Meyers

KU Leuven, Department of Mechanical Engineering, Celestijnenlaan 300A, B3001 Leuven, Belgium.

*Correspondence to:* Johan Meyers (johan.meyers@kuleuven.be)

**Abstract.** Given a wind-farm with known dimensions and number of wind-turbines, we try to find the optimum positioning of wind-turbines that maximises wind-farm energy production. In practise, given that optimisation has to be performed for many wind directions and taking into account the yearly wind distribution, such an optimisation is computationally only feasible using fast engineering wake models such as, e.g., the Jensen model. These models are known to have accuracy issues, in particular since their representation of wake interaction is very simple. In the present work, we propose an optimisation approach that is based on a hybrid combination of Large-Eddy Simulations (LES) and the Jensen model, in which optimisation is mainly performed using the Jensen model, and LES is used at a few points only during optimisation for online tuning of the wake-expansion coefficient in the Jensen model, and for validation of the results. An optimisation case study is considered, in which the placement of 30 turbines in a 4 by 3 km rectangular domain is optimised in a neutral atmospheric boundary layer. Both optimisation for single wind direction, and multiple wind directions are discussed.

## 1 Introduction

Wind-turbines are often clustered together in wind-farms, to save the cost of land and cabling. However, aerodynamic interactions between the turbines in the form of so called wakes (low speed regions) that form behind wind-turbines, lead to power reductions in 'waked' turbines of up to $50\%$ compared to a lone standing wind-turbine in undisturbed flow (Barthelmie et al., 2010). These interactions are very important when considering the topological placement of wind-turbines in large wind-farms.

In order to optimally design wind-farm layout, models are necessary that accurately predict the aerodynamic turbine-wake interaction effects. Such models need to be very fast, as wind-farm design optimisation needs to consider the full spectrum of wind directions over a wind-farm's operational lifetime, thus requiring many thousands of model evaluations. Moreover, wind-farm design is a multidisciplinary problem in which the aerodynamic wake-interaction model is only one of the models, next to turbine load models, lifetime analysis, economic investment models, etc. (see, e.g. Zaaijer, 2013). Today, the wake model that is most used is the Jensen model (Jensen, 1983; Katic et al., 1986). It is a simple and fast model, but is known to be inaccurate when looking at individual power predictions of turbines in various waked conditions (Barthelmie et al., 2009; Gaumond et al., 2014; Niayifar and Porté-Agel, 2015). Layout optimisation of wind-farms using fast wake models has been investigated in numerous studies (Marmidis et al., 2008; Emami and Noghreh, 2010; Kusiak and Song, 2010; González et al., 2010; Saavedra-Moreno et al., 2011; Du Pont and Cagan, 2012; Chowdhury et al., 2012; Samorani, 2013; Chen et al., 2013b).

However, the accuracy of such optimisation results has always remained a concern in view of the limited reliability of wake models, and this has recently led to a renewed interest in the formulation of accurate, but fast wake models (Stevens et al., 2015; Niayifar and Porté-Agel, 2015).

In the last five years, the detailed simulation of wind-farm atmospheric-boundary-layer interaction and turbine-wake interactions based on high-fidelity simulation tools such as Large-Eddy Simulations (LES) have become very popular (see, e.g., Meyers and Meneveau, 2010; Calaf et al., 2010; Yang et al., 2012; Meyers and Meneveau, 2013; Wu and Porté-Agel, 2013; Allaerts and Meyers, 2015), leading to many new insights in the flow physics of wind-farms. Given known and constant meteorological conditions, these type of models provide a detailed time-resolved prediction of the turbulent flow in a wind-farm with resolution of spatial flow structures in the order of 20 meters, and temporal fluctuations in the order of 10 seconds. Although it is computationally infeasible in LES of wind-farms to resolve all the detailed flow physics, such as, e.g., the turbine blade-boundary layers (with length scale below a millimeter), these models do lead to quite accurate predictions of wakes and wake merging, when compared to wind-tunnel and field experiments (Porté-Agel et al., 2011; Wu and Porté-Agel, 2013; Munters et al., 2016a). Unfortunately, LES of wind-farms requires supercomputing, and simulation times that are several hours to days for one single atmospheric condition. Hence, these models are not useful for layout optimisation purposes.

In the current work, we investigate a hybrid approach in which the Jensen model is used during optimisation, but we use LES to gradually adapt the Jensen Model, and verify the optimisations results. To this end, the wake-expansion coefficient in the Jensen model is iteratively fitted based on LES. In itself, tuning of the wake-expansion coefficient, e.g., to experiments, is quite common, but it is well-known that the coefficient depends on atmospheric conditions, farm layout, and may also best depend on streamwise distance into the farm (Stevens et al., 2015). Therefore, a coefficient that is tuned a priori will not fit all possible scenarios that are encountered during layout optimisation of a wind-farm over its relevant range of atmospheric conditions. In a hybrid Jensen–LES approach, it is possible to adapt the coefficient a posteriori during optimization depending on layout, wind direction, etc. The main focus of the current work, is on the formulation of an approach that is computationally feasible, given the very high costs of performing LES (even in a hybrid Jensen–LES optimisation). We demonstrate the proposed methodology on a moderately sized wind farm of 30 turbines in a 4 by 3 km farm area.

This paper is organised as follows. In Section 2 the mathematical formulation for the optimisation problem is stated, and the simulation models (both Jensen and LES), and the optimisation methodology are introduced. In Section 3, results are presented. First, the different steps in the algorithm are highlighted for a single wind direction optimisation case in §3.1–§3.3. Subsequently, in §3.4, some results for optimisation with multiple wind directions are discussed. Finally, conclusions are presented in Section 4

## 2  Problem Description and Methodology

In §2.1, the optimisation problem description is introduced. Subsequently, the Jensen model is briefly reviewed in §2.2. The LES simulation environment is discussed in §2.3, and finally the hybrid Jensen–LES approach and the optimisation method are presented in §2.4.

## 2.1 Problem Description

Consider a set of $N_t$ turbines that are to be placed in a fixed domain $\Omega$. Given constant atmospheric conditions and wind direction (parameterized in a vector $\boldsymbol{\mu}$), the average power output of a turbine at position $\boldsymbol{x}_i$ in the wind-farm is:

$$\overline{P}_i(\boldsymbol{x}_i, \boldsymbol{\mu}) = \frac{1}{T} \int_0^T P_i(\boldsymbol{x}_i, t, \boldsymbol{\mu}) \, \mathrm{d}t, \tag{1}$$

where $P_i(\boldsymbol{x}_i, t, \boldsymbol{\mu})$ corresponds to the instantaneous power output of the turbine (given atmospheric conditions $\boldsymbol{\mu}$), which is subject to turbulent wind fluctuations, and $T$ is a time averaging window that is sufficiently long to average out the turbulence effects. Note that the Jenson model (cf. §2.2) directly predicts $\overline{P}_i$ of turbines in a wind-farm, while, e.g., experimental measurements, as well as results from LES (cf. §2.3) yield $P_i(\boldsymbol{x}_i, t, \boldsymbol{\mu})$, and thus explicitly require above time averaging.

The optimisation problem that we consider is formulated as follows

$$
\begin{aligned}
\underset{\boldsymbol{x}_i}{\text{maximise}} \quad & \int \sum_{i=1}^{N_t} \overline{P}_i(\boldsymbol{x}_i, \boldsymbol{\mu}) f_p(\boldsymbol{\mu}) \, \mathrm{d}\boldsymbol{\mu} \\
\text{subject to} \quad & \boldsymbol{x}_i \in \Omega, \quad \forall i \in \{1, \cdots, N_t\} \\
& \|\boldsymbol{x}_i - \boldsymbol{x}_j\|_2 \geq d_{min}, \quad \forall i,j \in \{1, \cdots, N_t\}, \ i \neq j
\end{aligned}
\tag{2}
$$

where $\Omega$ is the wind-farm domain in which turbines can be freely placed, and $f_p(\boldsymbol{\mu})$ is the joint probability density function of atmospheric conditions $\boldsymbol{\mu}$ over which needs to be optimised (e.g. the yearly wind-direction distribution, atmospheric stability class, etc.). Finally, $d_{min}$ is a constraint on the minimum distance between turbines. In theory, the minimum distance between turbines is $1.0D$ (with $D$ the rotor diameter). In the current study, we will consider a minimum distance $d_{min} = 2.0D$ for all optimisation cases.

The solution of above optimisation problem requires a model for $\overline{P}_i(\boldsymbol{x}_i, \boldsymbol{\mu})$. This is discussed next in §2.2 for the Jensen model, and §2.3 for the LES model. To solve the above optimisation problem, we use the cross-entropy optimisation method (De Boer et al., 2005; Rubinstein, 1999) in combination with a hybrid Jensen–LES model as discussed in §2.4. Finally, note that for ease of notation, we drop $\boldsymbol{\mu}$ as an argument in $\overline{P}_i$. In fact, the conditions $\boldsymbol{\mu}$ (e.g. wind direction, turbulence intensity, etc.) are implicity contained on the set-up and boundary conditions of the respective models below.

## 2.2 The Jensen wake model

We briefly review the Jensen wake model as originally developed by Jensen (1983); Katic et al. (1986).

The model commences by assuming that each turbine generates a radially and azimuthally uniform wake that linearly expands with downstream distance from the turbine. Using simple mass conservation, this allows to describe the velocity deficit generated by turbine $i$ as

$$\Delta U_i(s_i) = U_\infty \frac{1 - \sqrt{1 - C_{T,i}}}{(1 + k_w s_i / R)^2}, \qquad s_i > 0, \tag{3}$$

with $C_{T,i}$ the turbine thrust coefficient, and where $s_i = (\boldsymbol{x} - \boldsymbol{x}_i) \cdot \boldsymbol{e}_f$ is the downstream axial distance from the turbine, and $\boldsymbol{e}_f$ the unit vector in the mean-flow direction. Obviously, $s_i > 0$. Upstream of a turbine, its own generated wake has a velocity deficit $\Delta U_i = 0$. Furthermore, $k_w$ is the linear wake-expansion coefficient, and $R$ is the rotor radius. Correlations exist that relate $k_w$ to the incoming atmospheric boundary layer, e.g. (Lissaman, 1979; Frandsen, 1992),

$$k_w = \frac{u_*}{U_\infty} = \frac{\kappa}{\ln(z_h/z_0)}, \tag{4}$$

is commonly used, with $\kappa$ the Von Kàrmàn constant, $z_h$ the turbine hub height, and $z_0$ and $u_*$ the surface roughness and friction velocity of the incoming atmospheric boundary layer. Note that in the current study, we will use LES to adapt $k_w$ in our optimisation procedure as discussed in §2.4. Finally note that the wake expansion is vertically restricted by the ground once the wake radius grows larger than the turbine hub height. However, the ground is not directly modelled, but instead mirror turbines are added below the ground, with wakes that are included in the wake merging model (Lissaman, 1979) (cf. below).

In order to estimate the power output $\overline{P}_i$, the turbine's incoming mean velocity is required. It is modeled as $U_{i,in} = U_\infty - \Delta U_{i,in}$, with $U_\infty$ the wind-farm inflow velocity at hub height, and $\Delta U_{i,in}$ upstream velocity deficit experienced by turbine $i$. The deficit $\Delta U_{i,in}$ is heuristically modelled by quadratically adding upstream wake deficits as follows

$$\Delta U_{i,in} = \left[ \sum_{j \in \mathcal{S}_i} (\Delta U_j(s_{ij}))^2 \right]^{1/2}. \tag{5}$$

Here $\mathcal{S}_i$ is the set of all upstream turbines that have a wake that geometrically intersects with turbine $i$, and $s_{ij}$ is the distance along the wind direction between turbine $i$ and $j$. In order to include the effect of the ground on wake development, mirror turbines (below the ground) are added to the set $\mathcal{S}_i$ for each turbine whose wake is restricted by the ground. It is furthermore possible that wakes only partially overlap, in which case the rotor area of the inflow turbine is split into regions with different overlaps. More details on the approach can be found in Rathmann et al. (2007).

Once the turbine inflow velocities $U_{i,in}$ are determined, the power per turbine is calculated as:

$$\overline{P}_i(\boldsymbol{x}_i) = \frac{1}{2} C_{P,i} \rho U_{i,in}^3, \tag{6}$$

where $C_{P,i}$ is the wind-turbine's power coefficient. For an ideal turbine, $C_{P,i}$ follows from axial momentum theory from, i.e.

$$C_{P,i} = \frac{1}{2} C_{T,i} [1 - (1 - C_{T,i})^{1/2}]. \tag{7}$$

For a real turbine, $C_{P,i}$ can be expressed as function of $C_{T,i}$ and wind speed, either using a mapping specific to the turbine, or blade-element momentum theory, and this can be straightforwardly used in the Jensen model. In the current study, we will simply use above ideal relationship, as our main focus is on the development and demonstration of the hybrid Jensen–LES approach, and not so much on the specifics of the selected turbine model.

### 2.3 Large-eddy simulation Environment and Simulation Setup

Simulations are performed using SP-Wind, developed at KU Leuven (Meyers and Meneveau, 2010, 2013; Allaerts and Meyers, 2015; Goit and Meyers, 2015; Munters et al., 2016a). SP-Wind solves the filtered incompressible Navier–Stokes equations,

which are given by

$$\nabla \cdot \widetilde{\boldsymbol{u}} = 0 \tag{8}$$

$$\frac{\partial \widetilde{\boldsymbol{u}}}{\partial t} + \widetilde{\boldsymbol{u}} \cdot \nabla \widetilde{\boldsymbol{u}} = -\frac{1}{\rho} \nabla \widetilde{p} + \nabla \cdot \boldsymbol{\tau}_M - \boldsymbol{f} \tag{9}$$

where $\widetilde{\boldsymbol{u}}(\boldsymbol{x}, t) = [\widetilde{u}_1, \widetilde{u}_2, \widetilde{u}_3]$ is the resolved velocity field, $\widetilde{p}$ is the pressure field, and $\boldsymbol{\tau}_M$ is the subgrid-scale (SGS) model. We use a standard Smagorinsky model (Smagorinsky, 1963) with Mason & Thomson's wall damping (Mason and Thomson, 1992) to model the SGS stress. Furthermore, $-\boldsymbol{f}$ represents the forces (per unit mass) introduced by the turbines on the flow. In LES of wind-farm boundary layers, this turbine-induced force is commonly modelled using an actuator-disk model (ADM), as full meshing of the turbine blades and geometry leads to computational grids that are too large for current-day computers. Expressed for turbine $i$, this force corresponds to (Meyers and Meneveau, 2010; Goit and Meyers, 2015):

$$\boldsymbol{f}^{(i)} = \frac{1}{2} C'_{T,i} \widehat{V}_i^2 \mathscr{R}_i(\boldsymbol{x}) \boldsymbol{e}_\perp \quad i = 1 \cdots N_t, \tag{10}$$

where $\boldsymbol{e}_\perp$ represents the unit vector perpendicular to the turbine disk, and $\mathscr{R}_i(\boldsymbol{x})$ is a geometrical smoothing function that distributes the uniform surface force of the turbine over surrounding LES grid cells, with $\int_\Omega \mathscr{R}_i(\boldsymbol{x}) \mathrm{d}\boldsymbol{x}' = A$ and $A$ the turbine disk area. Moreover, $\widehat{V}_i$ is the disk-averaged turbine velocity, and $C'_{T,i}$ is the disk-based thrust coefficient. Unlike the conventional thrust coefficient $C_T$ (used in the Jensen model) which is based on undisturbed velocity far upstream of a turbine, $C'_{T,i}$ is defined using the velocity at the turbine-disk. It results from integrating lift and drag coefficients over the turbine blades, taking design geometry and flow angles into account (cf. Appendix A in Goit and Meyers, 2015 for a detailed formulation). Based on axial momentum theory, we have (Calaf et al., 2010):

$$C_T = \frac{C'_T}{(1 + C'_T/4)^2} \tag{11}$$

which provides a direct relation between the thrust coefficient used in the Jensen model, and the disk-based thrust coefficient used in the LES model. Finally, given the velocity field $\widetilde{\boldsymbol{u}}(\boldsymbol{x}, t)$ from a LES, the average power output for turbine $i$ is determined from

$$\overline{P}_i(\boldsymbol{x}_i) = \frac{1}{T} \int_0^T \iiint \boldsymbol{f}^{(i)} \cdot \widetilde{\boldsymbol{u}} \, \mathrm{d}\boldsymbol{x} \, \mathrm{d}t. \tag{12}$$

In Figure 1 a typical snapshot of a horizontal velocity field $\widetilde{u}_1(\boldsymbol{x}, t)$ is shown, including an outline of the simulation domain that is considered in the current study. The main domain size is $L_y \times L_x \times L_z = 8.0 \times 6.0 \times 1.0$ km$^3$, where $x$ is always the main flow direction, and $z$ is the vertical direction. The wind-farm is inserted in a subdomain $\Omega = 4.0$km $\times 3.0$km (also marked on the figure). At $z = 0$ a classical high-Reynolds-number wall-stress boundary condition is used (Moeng, 1984; Bou-Zeid et al., 2005), which is parametrized by the ground surface roughness $z_0$, for which we use $z_0 = 0.1$ m. At $z = L_z$ a symmetry condition is used, and in the $y$ direction, periodic boundary conditions are used. Finally, at $x = 0$ an inflow boundary condition is used.

The inflow is generated in a separate precursor simulation (also shown in Figure 1), which employs shifted periodic boundary conditions to avoid artificial spanwise locking of the typical low-speed streaks observed in boundary layers (cf. Munters et al.,

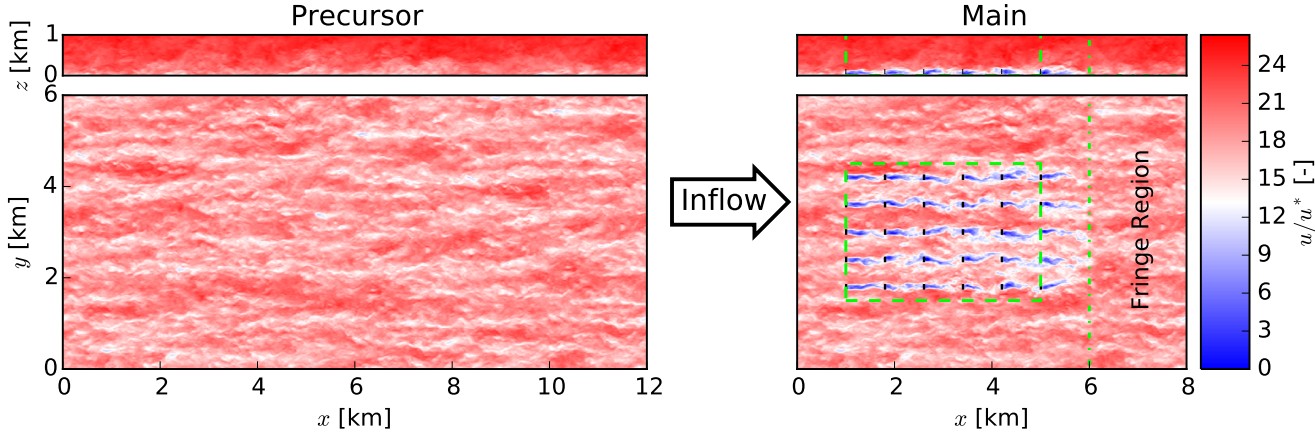

**Figure 1.** Snapshot of an instantaneous velocity field in the precursor domain and main simulation domain obtained from SP-Wind. Side view (top), plan view (bottom); precursor (left), main (right). Wind-farm area $\Omega$ (green dashed); Fringe region (green dash-dot).

2016b for details). For the precursor simulation, a domain size of $8.0 \times 6.0 \times 1.0$ km$^3$ is selected. The precursor simulation is driven by a constant pressure gradient, which corresponds to $\nabla p_\infty / \rho = u_*^2 / L_z$, where $u_* = (\tau_w/\rho)^{1/2}$ is the friction velocity in the precursor domain. In the current work, we are interested in Region II operation of wind turbines for which $C_T'$ can be presumed to be constant. Given that also $z_0$, and $z_h$ are fixed, simulation results remain dynamically equivalent for any selected

value of $u_*$, with velocity scaling proportionally with $u_*$, and time scaling inversely proportionally with $u_*$. An output of our precursor simulation (given $z_0 = 0.1m$) is the hub height velocity $u_h \approx 17.5 u_*$. Thus, to obtain a realistic region II hub-height velocity of, e.g., 8m/s, it suffices to select $u_* = 0.457$m/s. However, since all later results and comparisons with the Jensen model are normalized with inflow velocity, or with first-row power output, the exact value of $u_*$ is further not important (in our simulations, we just use $u_* = 1$).

For the discretization of the governing equations, SP-Wind uses a pseudo-spectral method in the horizontal directions, applying the $3/2$ rule for dealiasing (Canuto et al., 1988). In the vertical direction, a fourth-order energy-conservative finite-difference discretization scheme is used (Verstappen and Veldman, 2003). Non-periodic boundary conditions in the $x$-direction are implemented using a fringe-region technique, with a fringe region located in the last 2 km of the domain (see Spalart and Watmuff, 1993; Stevens et al., 2014; Munters et al., 2016b, a, for details). Mass is conserved by using a Poisson equation for

the pressure, that is solved using a direct solver. Finally, time integration is performed using a classical four-stage fourth-order Runge–Kutta scheme. For the simulations discussed in this paper, a fixed time step of $0.4$ seconds corresponding to a Courant–Friedrichs–Lewy (CFL) number of approximately 0.4 is used. The computational grid for the main domain corresponds to $N_y \times N_x \times N_Z = 256 \times 256 \times 80$, and $256 \times 256 \times 80$ in the precursor domain. For nonlinear operations we use the $3/2$ dealiasing rule, so that all nonlinear operations in real space are performed on $384 \times 384 \times 80$ grids for both domains. Simulation

parameters are summarised in Table 1.

**Table 1.** Simulation Parameters. Results remain dynamically equivalent for any selected value of the friction velocity $u_*$. The hub-height velocity obtained in the precursor simulation corresponds to $u_h \approx 17.5 u_*$.

| | |
|---|---|
| Total Domain Size (with Fringe Region) | $8 \times 6 \times 1$ km$^3$ |
| Total Domain Size (without Fringe Region) | $6 \times 6 \times 1$ km$^3$ |
| Optimisation Domain Size | $4 \times 3$ km$^2$ |
| Turbine Diameter | 100m |
| Turbine Height | 100m |
| Driving pressure gradient (precursor) | $-u_*^2/1000$ m/s$^2$ |
| Surface Roughness | 0.1m |
| Computational Reynolds No. | 100 |
| Grid Size | $256 \times 256 \times 80$ |
| Cell Size | $31.25 \times 23.44 \times 12.5$ m$^3$ |
| Time-step | $0.4/u_*$ seconds |

In the current study, we consider a rectangular fixed wind-farm domain $\Omega$ of 4.0 by 3.0 km (cf. above), in which thirty turbines are to be optimally placed. We take generic wind-turbines with a diameter of $D = 100$ m, and hub height of $z_h = 100$ m each. The selected disk-based and standard thrust coefficients correspond to $C_T' = 2.0$, and $C_T = 8/9$ respectively. The choice of turbines, simulation domain and selected computational grids correspond to the typical case set-ups found in Calaf et al. (2010);

Meyers and Meneveau (2013), and we refer the reader to these studies for detailed grid sensitivity analysis, etc.

Finally, simulations are initialized by first performing a spin-up of the turbulence in the precursor simulation. Starting from a logarithmic mean profile with random perturbations, the precursor simulation is advanced in time for $15000/u_*$ seconds so that realistic turbulence can develop. Subsequently, for every wind-farm layout, precursor and main domain are run simultaneously,

and an additional spin-up period of $2000/u_*$ seconds is simulated. This corresponds to at least $5u_*$ flow-through times of the main domain. At this point in time, time averaging of LES results is started.

In Figure 2, a detailed convergence analysis of the farm power and the power output of a single turbine is shown for an aligned wind-farm layout (corresponding to Case 4 in Table 2 below). In Figure 2a, a power histogram is shown for the wind farm, as well as for two individual turbines in the farm. Figure 2b shows results of the relative error $\epsilon_P$ of the time average as

function of the averaging time T (cf. Eq. 1), where

$$\epsilon_P(T) = \frac{\frac{1}{T}\int_0^T P(t)\,\mathrm{d}t - \overline{P}_{ref}}{\overline{P}_{ref}}. \tag{13}$$

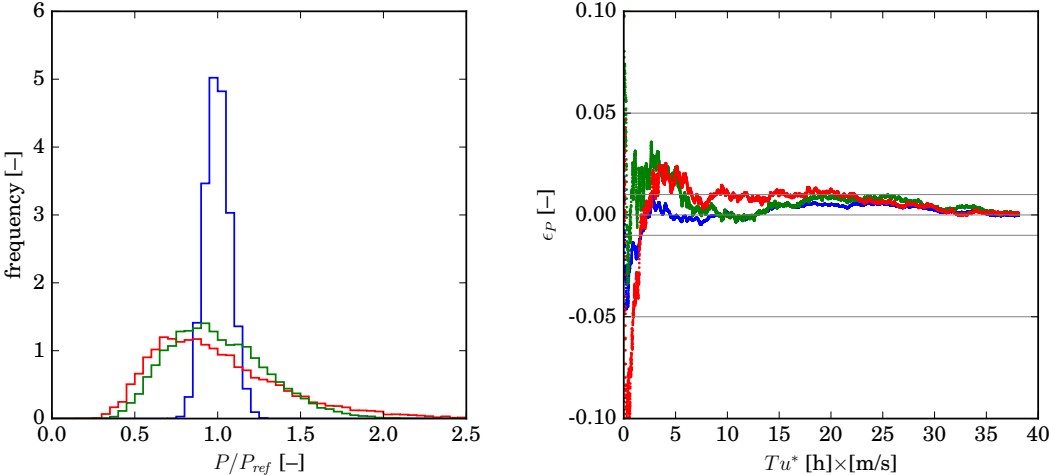

**Figure 2.** Convergence analysis of wind-farm and turbine power of an aligned wind-farm case (Case 4 in Table 2). (a) Pdf of wind-farm power output and power output of a front-row and backrow turbine. (b) Convergence error $\epsilon_P$ as function of averaging time $T$ for the wind-farm power, and for the power of a front-row and back-row turbine. (blue line): wind-farm power; (green line): first-row turbine; (red line): last-row turbine.

As reference $\overline{P}_{ref}$ we use an average obtained over a period of $40/u_*$ (taking $u_* = 0.457$m/s as a realistic value, this corresponds to averaging over 88 hours in physical time). It is appreciated from the figure that for limited averaging times, errors can be quite significant, in particular when looking at the single-turbine average. In fact, it is well known that the time average in turbulent flows converges as $T^{-1/2}$ (see,e.g., Tennekes and Lumley, 1972). This is also appreciated in Figure 2b: errors
decrease fast at low values of $T$, but afterwards convergence stagnates. This is particularly problematic when looking at the turbine average power, which requires roughly $15/u_*$ to $20/u_*$ hours to converge within 1 percent of the reference average (requiring excessive computational costs – cf. below). It is further appreciated that the error on the overall wind-farm power converges significantly faster, i.e. an error of 1 percent is reached after approximately $5/u_*$ hours. This is related to the fact that $N_t$ partly uncorrelated turbine power signals are accumulated. Therefore, in order to limit computational effort related to
LES in a hybrid Jensen–LES approach, we will formulate our approach based on matching LES and Jensen farm power levels. In order to avoid over fitting of the Jensen wake-expansion coefficient, we use an ensemble of different wind-farm layouts that gradually evolve during optimization towards layouts that are more optimal in terms of power extraction. This approach is further discussed in §2.4.

  In terms of computational cost, the spin-up of the precursor simulation is the most expensive (but needs to be done only
15 once), amounting to 32 hours of wall-clock time on the ThinKing cluster of the Flemish Super-Computer Centre, using 8 Ivy Bridge nodes consisting of two 10-core 'Ivy Bridge' Xeon E5-2680v2 CPUs (2.8 GHz, 25 MB level 3 cache) for a total of 160 cores. Wind-farm spin-up takes around 14 hours of wall-clock time on the same processor layout. Subsequent averaging takes around 9 hours of wall-clock time per $3600/u_*$ seconds of wind-farm time. In order to keep overall computational costs under

control, we limit time averaging in the current work to $3600/u_*$ seconds (roughly corresponding to at least $9u_*$ flow-through times). This yields an expected error level on the power output of 2% (cf. discussion above and Figure 2b). In practice, for optimization over a single atmospheric condition $\boldsymbol{\mu}$ (e.g. a single wind direction), it may be advisable to use at least $5/u_*$ for the current case set-up. However, when considering optimization over a range of conditions, the impact of this variability will

be further averaged out.

## 2.4   Hybrid Jensen–LES approach and cross-entropy optimization

In the current manuscript, we propose a hybrid Jensen–LES approach for wind-farm layout optimisation. To that end, the layout optimization is based on the Jensen model, but the wake expansion coefficient $k_w$ is iteratively used to fit the Jensen model to a set of LES data that is gradually adapted to the layouts that are explored during optimisation. The approach is summarized

in Algorithm 1. Here, we describe the approach considering a single atmospheric condition $\boldsymbol{\mu}$ (cf. Eq. 2), e.g., a single wind direction. Generalization is straightforward, and optimization over different wind directions will be discussed in §3.4.

In a first step, a set of $N_L$ LES cases of regular and random layouts are generated. This set is used to fit $k_w$ using Algorithm 3 (see below). Subsequently, layout optimisation is performed using the Jensen model and Algorithm 2 (see further below). The optimal layout is then added to the set of LES cases, and a number of $N_R - 1$ ($N_R < N_L$) additional random layouts are added

as well. Moreover, the $N_R$ LES cases with lowest generated powers are removed from the set. This new set is used to refit $k_w$, subsequently starting a new layout optimisation, etc. By doing so, the LES data set used for fitting is gradually taking more optimal layouts into account, while layouts that are least optimal are removed from the set.

The procedure described above, directly uses wind-farm power to fit the wake expansion coefficient, and avoids using errors on individual turbine power output. As discussed in §2.3, this reduces the need for time averaging in the LES, and significantly

lows computational costs. Moreover, by including $N_L$ different layouts, potential overfitting of $k_w$ is avoided, and the influence of remaining LES convergence errors on the optimal fit is further reduced.

For the layout optimization in Algorithm 2 and the optimal fit of $k_w$ in Algorithm 3, we employ the Cross Entropy (CE) method. This method was originally developed to estimate the probability of rare events. Later on, it was realised that it is also very effective in solving difficult non-convex optimisation problems. The method is explained in detail by De Boer et al.

(2005); Rubinstein (1999), among others. Here, we briefly review the main features of the approach, and further detail how we use it in a hybrid Jensen–LES optimisation of wind-farm layout. In our hybrid Jensen–LES optimisation approach of wind-farm layout, we use the CE method both for Jensen-based layout optimization, as well as for the adaptive fitting of the Jensen wake expansion coefficient against a range of LES results (as further detailed below). However, it is important to emphasize that any feasible optimization method may be used for this. For instance, recently, some work has focussed on the use of a gradient-

based layout optimization approach in combination with engineering wake models (Fleming et al., 2016), while before others have, e.g., looked into the use of a particle-swarm method (Wan et al., 2010), genetic algorithms (Chen et al., 2013a), etc.

First of all, the optimisation problem (2) is slightly reformulated in order to better cope with the second inequality constraint (as further discussed below, the first constraint is more straightforward to enforce directly). Therefore, we consider following

**Algorithm 1: Main Algorithm**. A summary of the overall procedure used to obtain the optimum wind-farm layout. Values used in the current study are $N_L = 10$, and $N_R = 5$.

**Input:** Dimensions $L_x$ and $L_y$ of the wind-farm domain, Number of wind-turbines $N_t$, Diameter of wind-turbines $D$, Minimum acceptable distance between the wind-turbines $d_{min}$

**Output:** Optimum value of wake expansion coefficient, An optimum wind-farm layout

Generate a set of $N_L$ LES cases for initial calibration of the Jensen model (choose aligned, staggered, and random layouts)

Using the available LES data and Algorithm 3, optimise the wake expansion coefficient for the Jensen Model by minimizing the error between LES and Jensen wind-farm powers over the different $N_L$ layouts

Using Algorithm 2 and the Jensen model, find the optimum wind-farm layout

Verify the optimisation results using LES

Add the optimum layout to the set of LES cases; add an additional set of $N_R - 1$ random layouts (that satisfy all constraints); remove $N_R$ $(< N_L)$ cases that have the lowest cost function from the LES data set

Repeat Steps 2 to 5 until the error between the LES and Jensen Model in the optimal layout is less than a pre-specified threshold

non-smooth problem

$$\max_{\boldsymbol{x}_i} \sum_{i=1}^{N_t} \overline{P}_i(\boldsymbol{x_i}) + \sum_{i=1}^{N_t} \sum_{j=1}^{i-1} h_{ij}(\boldsymbol{x}_i, \boldsymbol{x}_j) \tag{14}$$

s.t.

$$\boldsymbol{x}_i \in \Omega, \quad \forall i \in \{1, \cdots, N_t\}, \tag{15}$$

5 where

$$h_{ij}(\boldsymbol{x}_i, \boldsymbol{x}_j) = \begin{cases} -\infty & \|\boldsymbol{x}_i - \boldsymbol{x}_j\|_2 < d_{min} \\ 0 & \text{otherwise} \end{cases} \tag{16}$$

This formulation is fully equivalent to (2).

The CE method for solving the optimal placement problem now essentially involves three steps. In a first step, a set of $N_s$ uniformly distributed random samples of the optimisation parameters $\boldsymbol{x}_i$ are generated with a given mean value $\boldsymbol{m}^{(0)}$ and

10 deviation $\boldsymbol{d}^{(0)}$ (note that both $\boldsymbol{m}$ and $\boldsymbol{d}$ have dimension $2 \times N_t$). At startup (iteration 0), no prior knowledge of the optimisation problem is available, so we chose the mean and deviation such that the distribution spans the whole feasible parameter range $\Omega$. In the second step, samples are sorted according to their cost functional value. The best $N_b < N_s$ samples are chosen, and the mean $\boldsymbol{m}_b^{(k)}$ and deviation $\boldsymbol{d}_b^{(k)}$ of this set (in iteration step $k$) is calculated. In a third step, a next generation of samples (iteration step $k+1$) is then created using a uniform distribution with mean and deviation

$$\boldsymbol{m}^{(k+1)} = \boldsymbol{m}^{(k)} + \alpha(\boldsymbol{m}_b^{(k)} - \boldsymbol{m}^{(k)}) \tag{17}$$

$$\boldsymbol{d}^{(k+1)} = \boldsymbol{d}^{(k)} + \alpha(\boldsymbol{d}_b^{(k)} - \boldsymbol{d}^{(k)}) \tag{18}$$

where the parameter $\alpha$ is selected in the $[0,1]$ range, specifying how conservative or exploratory the algorithm is. This procedure continues until the end condition is met, which is usually set by specifying the maximum number of iterations. We also transfer the optimum value in each generation to the next generation, so that the cost function value of the optimum in each generation increases monotonically.

The treatment of the constraint $\boldsymbol{x}_i \in \Omega$ is straightforward. Whenever a turbine location in a sample falls outside $\Omega$, the location is simply orthogonally projected on the boundary of $\Omega$. Note that turbines in samples in the initial generation always fall in $\Omega$, but in later generations, this is not always the case. Though the projection on $\Omega$ will slightly change the distribution, as relatively more sample points can end up on the boundary, we did not find this to hamper the convergence of our algorithm. Finally, the treatment of the distance constraint is implicitly handled by the cost function formulation, and does in principle not
require any further attention.

Given the Jensen model, and an input for the wake expansion coefficient $k_w$, the cross-entropy layout optimisation is summarized in Algorithm 2, and specific choices are documented. We run the Cross Entropy Optimisation scheme for $N_{iter} = 2000$ iterations, however, we find it beneficial for convergence and computational efficiency to omit $h_{ij}$ in the cost function during the first $M$ iterations, and only enforce $h_{ij}$ constraint for $k > M$. We take $M = 200$ in our implementation.
The standard deviation of samples in the cross entropy scheme eventually converge to zero. Once the standard deviation has become small, and if the algorithm is locked in a local optimum, it will not anymore break away from it. To reduce the chance of this happening, we reset the calculated value of $\boldsymbol{d}$ after 1000 iterations. For turbines with $x$ coordinate less than $0.5$ km or bigger than $3.5$ KM, we reset their corresponding deviation to $[0.5, 0.5]$ and for the rest we reset the deviation to $[L_x/2, L_y/2]$. This can be interpreted as running the Cross Entropy in two stages. Both run for 1000 iterations: the first runs starting with
a uniform distribution in $\Omega$, and the second starts with the optimum layout of the first stage as the mean value for its initial population. In the interest of simplicity, this detail is not included in the outline of Algorithm 2.

A second algorithm that is used in Algorithm 1 is the fitting of the wake expansion coefficient $k_w$ to the LES data. Fitting $k_w$ is also a non-convex optimisation problem, and therefore, we simply use the CE method again, but now for a scalar field. This is summarized in Algorithm 3. For this fitting, we found a number of iterations $N_{iters} = 50$ sufficient for good convergence.
We remark here that Algorithm 3 can in principle be used to fit more complicated relations for $k_w$. For instance, introducing the heuristic dependence $k_w = a + bx$ (or similar expressions), and fitting $a$, and $b$ instead of the mean value of $k_w$, may be interesting approach to represent the downstream development of $k_w$ in the wind-farm related to increased turbulence levels. In the current work, we did not further explore this type of parametrizations of $k_w$, as a simple fit of the mean value already leads to very satisfactory results (cf. next section).
Finally, we remark that the CE method is a global optimization method. However, its convergence to the global optimum in a finite number of iterations can only be formally proven for some specific conditions, and in practice convergence depends on a number heuristic choices, and is difficult to formally prove (see, e.g., Rubinstein and Kroese, 2013, for details). In fact, this is a disadvantage that all global optimization methods share. However, the main advantage for using a global method is the fact that the algorithm does not get trapped in local optimums that easily. Moreover, the disadvantage of the high number of function
evaluations required for such global methods to work well is not really an issue, as Jensen-model evaluations are extremely

**Algorithm 2:** The outline of the Cross Entropy Optimisation Method for finding the optimum wind-farm layout. Values used in the current study are $N_{iters} = 2000$, $N_s = 1000$, $M = 200$.

---

**Input:** Value of the wake expansion parameter $k_w$, domain $\Omega$, minimum distance $d_{min}$.

**Output:** An optimum wind-farm layout that generates maximum amount of energy.

**1** **for** $k \leftarrow 1$ **to** $N_{iters}$ **do**

**2**     Generate $N_s$ random layouts, where each random sample consists of a set coordinates $\boldsymbol{x}_i$ $(i = 1 \cdots N_t)$. Samples are uniformly distributed with mean value of $\boldsymbol{m}^{k-1}$ and deviation of $\boldsymbol{d}^{k-1}$. Initial mean and deviation values are set to span the whole domain $\Omega$;

**3**     **if** $k > 1$ **then**

**4**         Replace the first sample with $S_{OPT}$;

**5**     **for** $j \leftarrow 1$ **to** $N_s$ **do**

**6**         Calulate the total power of layout $j$ (ommit $h_{ij}$ in cost function for $k \leq M$);

**7**     Sort the $N_s$ samples based on their total generated power in descending order.

**8**     Choose the best $N_b$ samples (we use $N_b = 0.4 N_s$).

**9**     Set $\boldsymbol{m}_b^{k-1}$ and $\boldsymbol{d}_b^{k-1}$ to be the mean and deviation of the best $N_b$ samples.

**10**     Calculate $\boldsymbol{m}^k$, and $\boldsymbol{d}^k$ using (17,18)

**11**     Set the best sample as the optimum layout $S_{OPT}$;

---

cheap. In fact, as further discussed below, the main cost in our overall hybrid method remains associated with performing the LES.

## 3 Results

In the current section, optimisation results are discussed. First of all, in §3.1, the initial LES database for calibration of the Jensen model is constructed. Next, optimisation results of the Jensen only model are discussed in §3.2. Subsequently, hybrid Jensen–LES optimisation results are presented in §3.3. Finally, optimisation for multiple wind directions is discussed in §3.4.

### 3.1 Set-up of LES database for initial calibration

A first step in Algorithm 1 is the generation of a LES database that is a starting point for the calibration of the Jensen model. Here we choose a mix of staggered, aligned layouts, and randomly generated layouts. An overview of the different cases, and their generated power is provided in Table 2. We normalize all results with the power output of a 'wakeless' wind-farm, i.e. a wind-farm consisting of turbines that all have undisturbed inflow. In order to normalize all LES results in the same way, we use the averaged power output of turbines located in the first row of the aligned and staggered layouts and multiply it by $N_t$ $(= 30)$ to find the 'wakeless' wind-farm output. We then state every wind-farm power output as a percentage of this 'wakeless' wind-

**Algorithm 3:** The outline of the Cross Entropy Optimisation Method for optimising the wake expansion coefficient in the Jensen Model using the LES data. Values used in the current study are $N_{iters} = 50$, $N = 1000$.

**Input:** Total power of $N_L$ wind-farm layouts obtained from LES simulations, each having $N_t$ wind-turbines.

**Output:** An optimum value for $k_w$ that minimises the error between predicted LES wind-farm power and Jensen wind-farm power.

Set the initial mean value $m^{(0)}$ and deviation $s^{(0)}$;

**for** $i \leftarrow 1$ **to** $N_{iters}$ **do**

Generate $N$ random scalar samples, with uniform distribution with mean value of $m^{(i-1)}$ and deviation of $s^{(i-1)}$;

**if** $i > 1$ **then**

Replace the first sample with $k_{w,OPT}$ ;

**for** $j \leftarrow 1$ **to** $N$ **do**

Using sample $j$ as $k_w$ in the Jensen Model, calculate the relative wind-farm power of $N_L$ layouts.

Define $e_j$ as the sum of absolute value of errors between the Jensen model and LES wind-farm powers for the $N_L$ layouts.

Sort the $N$ samples based on their error value $e_j$ for $j \in \{1, \cdots, N\}$, in ascending order.

Choose the first (best) $N_b$ samples (we normally set $N_b = 0.4N$).

Set $m^{(i)}$ and $s^{(i)}$ to be the mean and deviation of the best $N_b$ samples.

Set the best sample as the optimum value $k_{w,OPT}$.

**Table 2.** Large Eddy Simulation results for different wind-farm layouts. Power output normalized with respect to total power of a wind-farm consisting of 'first-row' turbines. Average LES power is 69.97%.

| Case No. | Description | Relative wind-farm Power |
|:---:|:---|:---:|
| 1 | Aligned with $5D \times 5D$ Spacing | 51.81% |
| 2 | Aligned with $6D \times 5D$ Spacing | 56.76% |
| 3 | Aligned with $7D \times 5D$ Spacing | 60.80% |
| 4 | Aligned with $8D \times 5D$ Spacing | 64.36% |
| 5 | Staggered with $8D \times 5D$ Spacing | 83.60% |
| 6 | Gradually staggered with $8D$ Spacing | 87.40% |
| 7 | Randomly generated with $d_{min} = 2D$ | 79.28% |
| 8 | Randomly generated with $d_{min} = 3D$ | 76.16% |
| 9 | Randomly generated with $d_{min} = 4D$ | 78.66% |
| 10 | Randomly generated with $d_{min} = 5D$ | 80.49% |

farm output. Looking at the results of Table 2 it is apparent that the aligned cases perform quite poor in terms of relative power output, considerably worse than the staggered cases, but also worse than any of the random layouts that we investigated.

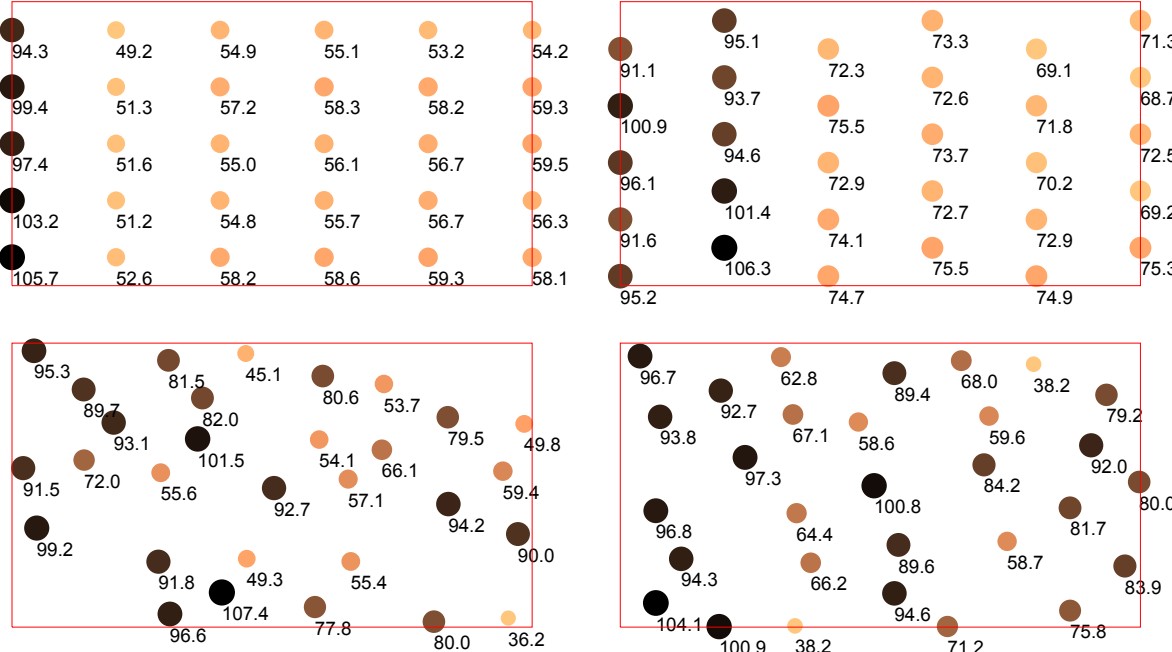

**Figure 3.** Layout and relative turbine power output for four of the cases listed in Table 2. Relative power results are obtained from large-eddy simulations. Turbine locations are marked with colored disk: size and color are scaled with relative power. Plot boundary (red line) corresponds to boundaries of domain $\Omega$ (cf. Figure 1).

In Figure 3 the layout and relative power output of individual turbines are shown for an aligned, staggered, and two of the random layouts. Wind direction is always from left to right. First of all, we remark that there is still considerable variability at turbine level that is due to the limited averaging period of 3600 seconds. As shown in Figure 2b, variability in the turbine power average is in the order of $\pm 5\%$ and more, and this is in line with the variability observed in the first row of Figure 3. We did verify that first-row turbine averages all converge to a relative power of 100% when averages of up to $15/u_*$ hours are used. Finally, we remind that the accumulated farm power is much better converged.

## 3.2 Comparison of Jensen model and LES results

Without access to reference results that can serve to tune $k_w$ in the Jensen model, it is possible to resort to Eq. 4 to determine $k_w$. Using this equation for our simulation set-up leads to

$$k_w = \frac{0.41}{\ln(100/0.1)} = 0.060$$

Here we briefly compare the Jensen model using this value with LES results. To do so, we use the 10 layouts presented in Table 2.

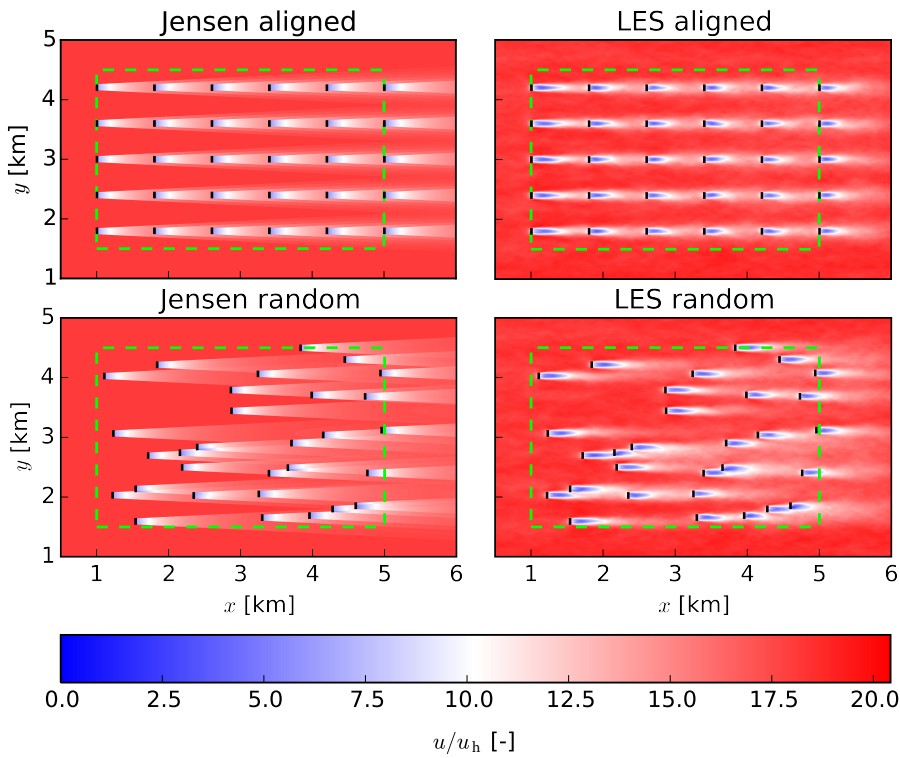

**Figure 4.** Comparison of Jensen Model and LES for an aligned and random layout.

A comparison of flow fields as generated by the Jensen model and LES are shown in Figure 4. It is appreciated that the averaged flow data of LES are much smoother as a result of turbulent mixing. In contrast, in the Jensen model wakes have a sharp boundary, also leading to sharply marked overlap regions. Note that also mirror wakes pop-up more downstream in the farm. Some features are not represented at all by the Jensen model. For instance, in the random layout, it is appreciated that
5    side-by-side wakes can influence each other. Such behaviour is not parametrized in the Jensen model.

However, the most relevant property from a power optimisation point of view is the total error on the predicted power. In table 3, the average power output from LES and the Jensen model are compared. It is appreciated that the Jensen model using $k_w = 0.060$ is very accurate for some cases, but not so for others. In particular, the cases that have a higher relative power extraction are generally predicted worse by the Jensen model, than the cases with a lower relative power (the most prominent
10    exception is Case 6). Another trend is that the regular cases are better predicted than the irregular cases. In the context of optimisation, it is however not important for the Jensen model to be accurate over a wide range of different layouts. Far away from the optimal layout, the required accuracy can be allowed to be considerably lower than close to the optimum. In this sense, Algorithm 1 gradually adapts the Jensen model through its wake expansion coefficient to better fit more performing layouts.

**Table 3.** Comparing outputs of LES and Jensen Wake Model with $k_w = 0.060$.

| Case | Relative Power (LES) | Relative Power (Jensen Model) | Error |
| --- | --- | --- | --- |
| Aligned $5D \times 6D$ | 51.21% | 52.30% | $-1.09\%$ |
| Aligned $6D \times 6D$ | 55.93% | 57.98% | $-2.05\%$ |
| Aligned $7D \times 6D$ | 60.13% | 62.88% | $-2.75\%$ |
| Aligned $8D \times 6D$ | 63.34% | 66.83% | $-3.50\%$ |
| Staggered $8D$ | 82.33% | 86.81% | $-4.48\%$ |
| Gradual Staggered $8D$ | 85.77% | 89.18% | $-3.41\%$ |
| Random1 | 78.29% | 85.20% | $-6.91\%$ |
| Random2 | 74.77% | 82.30% | $-7.53\%$ |
| Random3 | 77.96% | 84.95% | $-6.99\%$ |
| Random4 | 79.17% | 84.04% | $-4.87\%$ |

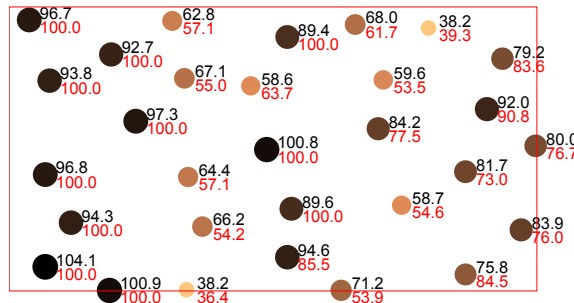

**Figure 5.** Comparing the wind-turbine power generation obtained from LES data (black font) and Jensen Model (red font). Turbine locations are marked with colored disk: size and color are scaled with relative power. Plot boundary (red line) corresponds to boundaries of domain $\Omega$ (cf. Figure 1).

Finally, looking at turbine level in Figure 5 for one of the random layouts (i.e. Case 10), it is seen that errors at turbine level are much bigger than the error on the accumulated power reported in Table 3. Again, from an optimisation point of view, this is less of an issue, as long as coupled approach in combination with LES is used to adapt the model and verify the overall results close to the optimum. We further notice here that the statistical errors on the averaged turbine power output from LES are still significant due to the limited time of averaging (in the order of $5\%$ – cf. discussion in §3.1).

### 3.3 Hybrid Jensen–LES optimisation

Using Algorithm 1, we now optimize the wind-farm layout with a single constant wind direction given the set-up in Figure 1, and wind coming from the left. Strictly speaking, this corresponds to the situation where $f_p(\boldsymbol{\mu})$ in (2) corresponds to a Dirac delta function centered around an eastern wind direction, so that the integral over atmospheric conditions in (2) drops out. Optimization over different wind directions is briefly discussed in §3.4. For the single wind-direction case considered here, only three outer iterations are required in the algorithm to converge to an optimal layout and optimally tuned Jensen model. Intermediate results of these iterations are discussed below.

**Iteration 1:** We start Algorithm 1 with the initial cases shown in Table 2. Using these 10 cases, we use Algorithm 3 to optimise the value of $k_w$ finding a value of $k_w = 0.055$. Subsequently, this value is used to optimize the layout using Algorithm 2. The resulting optimal layout is shown in Figure 6. Table 4 summarizes the relative LES and Jensen power, and errors for the 10 initial training cases, as well as for the newly obtained optimal layout. The relative power generated by the newly found optimum corresponds to $90.5\%$ (evaluated using the LES), but the error with the Jensen model is still noticeable, i.e. $-2.62\%$.

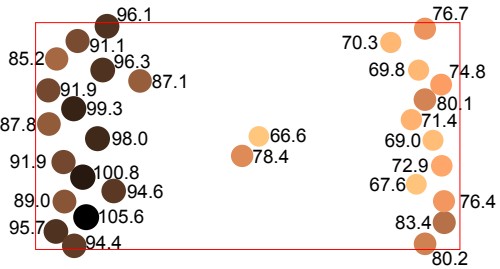

**Figure 6.** Optimal layout and relative power for a single wind direction obtained after Iteration 1. Relative power results are obtained from large-eddy simulations. Turbine locations are marked with colored disk: size and color are scaled with relative power. Plot boundary (red line) corresponds to boundaries of domain $\Omega$ (cf. Figure 1).

**Iteration 2:** We add optimal layout 1 and four additional random layouts to the LES database, and remove the 5 layouts with lowest relative power. Using Algorithm 3, we find a new value $k_w = 0.036$ that best fits the Jensen model to the LES data. Subsequently, using Algorithm 2, a new optimal layout is found, which is shown in Figure 7. Furthermore, an overview of relative powers from LES and Jensen is shown in Table 5. It is appreciated that the new optimal layout leads to a relative power of 92.8% (evaluated using LES), but in contrast to the first iteration, the error with the Jensen model remains now limited to 0.17%.

As can be seen, the two optimum layouts, although obtained using different values of $k_w$, have the same general structure.

**Iteration 3:** We repeat the procedure a third time, and find (almost) the same value for $k_w$. Only the fourth digit differs and the resulting new optimal layout remains the same. In fact, we observed that up to changes in the second digit, the value of

**Table 4.** Iteration 1: Comparing outputs of LES and Jensen Wake Model with $k_w = 0.055$.

| Case No. | Relative Power (LES) | Relative Power (Jensen Model) | Error |
|---|---|---|---|
| Aligned $5D \times 6D$ | 51.21% | 49.59% | 1.62% |
| Aligned $6D \times 6D$ | 55.93% | 55.40% | 0.53% |
| Aligned $7D \times 6D$ | 60.13% | 60.17% | $-0.04\%$ |
| Aligned $8D \times 6D$ | 63.34% | 64.22% | $-0.88\%$ |
| Staggered $8D$ | 82.33% | 85.78% | $-3.46\%$ |
| Gradual Staggered $8D$ | 85.77% | 92.27% | $-6.50\%$ |
| Random1 | 78.29% | 84.52% | $-6.23\%$ |
| Random2 | 74.77% | 81.23% | $-6.46\%$ |
| Random3 | 77.96% | 84.51% | $-6.55\%$ |
| Random4 | 79.17% | 83.27% | $-4.10\%$ |
| Optimum iter. 1 | 90.51% | 93.13% | $-2.62\%$ |

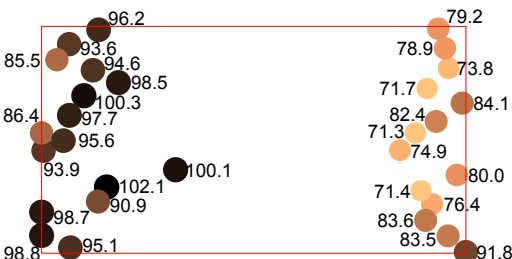

**Figure 7.** Optimal layout and relative power for a single wind direction obtained after Iteration 2. Relative power results are obtained from large-eddy simulations. Turbine locations are marked with colored disk: size and color are scaled with relative power. Plot boundary (red line) corresponds to boundaries of domain $\Omega$ (cf. Figure 1).

$k_w$ does not significantly influence the optimal layout. Finally, the error between the Jensen model and the LES is below 1%, which corresponds roughly to the statistical averaging accuracy of the LES. We conclude that the algorithm is converged.

After initial set-up of the LES database, each main optimization step requires $2.5 \times 10^6$ Jensen evaluations per iteration and 5 LES evaluations. Wall time for the Jensen evaluations (per iteration) corresponds roughly to $1, 25$ hours on 1 Ivy Bridge node of the ThinKing cluster of the Flemish Super-Computer Centre. Total wall time for LES (per iteration, and excluding the precursor spin-up time – cf. §2.3) amounts to approximately 70 hours on 8 nodes of the Flemish Super-Computer, equivalent

**Table 5.** Iteration 2: Comparing outputs of LES and Jensen Wake Model with $k_w = 0.036$.

| Case No. | Relative Power (LES) | Relative Power (Jensen Model) | Error |
|---|---|---|---|
| Staggered $8D$ | 82.33% | 79.01% | 3.32% |
| Gradual Staggered $8D$ | 85.77% | 93.06% | $-7.29\%$ |
| Random1 | 78.29% | 81.81% | $-3.52\%$ |
| Random3 | 77.96% | 82.27% | $-4.31\%$ |
| Random4 | 79.17% | 77.80% | 1.37% |
| Random5 | 79.54% | 82.06% | $-2.52\%$ |
| Random6 | 76.01% | 78.67% | $-2.65\%$ |
| Random7 | 80.96% | 83.50% | $-2.54\%$ |
| Random8 | 76.25% | 78.04% | $-1.79\%$ |
| Optimum iter. 1 | 90.51% | 90.17% | 0.34% |
| Optimum iter. 2 | 92.04% | 91.88% | 0.17% |

**Table 6.** Optimum values of $k_w$ obtained in different iterations of Algorithm 1

| Iteration No. | optimum $k_w$ | Relative LES Power of correspnding optimum layout |
|---|---|---|
| 1 | 0.055% | 90.51% |
| 2 | 0.036% | 92.04% |
| 3 | 0.036% | N/A |

to 560 node-hours. Even though the Jensen model is 500000 times more evaluated per iteration than LES, the total LES cost is roughly 500 times more expensive, and the LES wall time roughly 50 times longer.

Given this single wind direction, the optimal layout leads to a relative wind-farm performance of around 93% of a wakeless windfarm which is considerably higher than a typical aligned or staggered layout. Moreover, looking at the layout that was found in Figure 7, it is observed that turbines are grouped into two main clusters, one at the front of the farm, and one at the back of the farm, leaving a large stream-wise distance in between for wake recovery. Obviously, this result is particular for a single wind direction. In the next section, we study the cases with multiple wind directions.

Finally, we remark that it is difficult to prove formal convergence of the CE method that we use for our optimization (cf. discussion at the end of §2.4), and optimization is terminated based on a maximum number of iterations in Algorithms 2 and 3. Therefore, we checked the dependence of our results versus the initial starting point of the optimization. We found that a

change in initial distribution leads to slight shifts in the turbine locations, but this does not significantly influence the value of the power extraction. Moreover, we also experimented with the use of gradient-based optimization using the CE optimum as a starting point of the gradient-based method. To this end, we employed Matlab's fmincon routine. Unfortunately, including all nonlinear distance constraints did not work (given 30 turbines, there are 435 distance constraints). Omitting these in the gradient-based method, we found that turbine locations again slightly shift, but that power increases by 0.4% only, indicating the obtained CE optimum is well converged. Overall, we find that the cost function is relatively flat near the region of optimal power production, i.e. small shifts of the turbine locations do not lead to significant changes in power output.

### 3.4 Optimization for multiple wind directions

We now consider optimisation over a wind-direction distribution. Two cases are considered. The first corresponds to a uniform wind distribution over an angle of $\pm 7.5°$, representing a case with a dominant wind direction. The second corresponds to a uniform wind distribution over an angle of $\pm 180°$, representing a case without a dominant wind direction.

In order to properly represent power output over the wind distribution using the Jensen model, we sample the uniform distributions in $1.5°$ increments, and the integral in (2) is discretized using a Riemann sum based on these intervals each with constant probability. For LES evaluations, we use a much coarser sampling: for the dominant wind-direction case only 3, and for the uniform $360°$ case we use 8 directions. The error between Jensen model and LES is only defined relying on these distinct directions. In this way, the overall computational costs related to LES remains limited compared to the additional Jensen model evaluations that are performed.

We first focus on the dominant wind direction case, and perform an optimisation using the Jensen model, and $k_w = 0.036$ obtained in previous section. An overview of the errors between Jensen model and LES for the optimal layout is given in Table 7. It is appreciated that errors are already below $2\%$ for all directions, and therefore we do not further perform iterations using Algorithm 1 here. The overall optimal power output corresponds to $93.67\%$, and the related layout is shown in Figure 8. It is appreciated that the optimal layout for the dominant wind direction very much resembles the layout for the single wind direction case. Turbines are again clustered in two large groups, one in front and one at the back of the wind farm.

Finally, we look at optimisation for the uniform wind distribution. Again we perform optimisation using the Jensen model and $k_w = 0.036$. An overview of the errors for the optimal layout is provided in Table 8. Also now, errors are overall relatively low, so that for sake of saving computational resources, we do not perform further iterations using Algorithm 1. We further find that overall, the average power output of the optimised layout corresponds to $93.45\%$. This compares to $71.73\%$, and $75.25\%$ for the aligned $8D \times 6D$ and for staggered layout respectively.

The optimal layout itself is shown in Figure 9. In contrast to the layout found for the dominant wind direction, now turbines are spread out much more evenly throughout the domain. Moreover, a number of turbines, i.e. seven, are located on the domain boundary. We remark here, that for similar optimization cases in literature, turbines sometimes end up at the domain corners (see, e.g., Réthoré et al., 2014, or Feng and Shen, 2015), but this is not the case for all studies (e.g. Du Pont and Cagan, 2012). Currently, we are not sure whether this is possibly related to domain shape, size, and number of turbines, or whether this is related to the existence of local optimums, or convergence of the optimization method. Using a hybrid method that

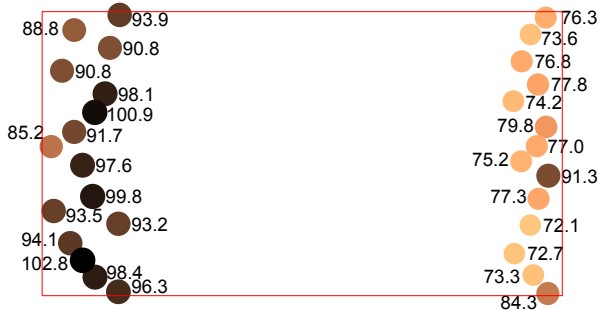

**Figure 8.** Optimal layout and relative power for dominant wind-direction case (angle of $\pm 7.5°$). Relative power results are obtained from large-eddy simulations. Turbine locations are marked with colored disk: size and color are scaled with relative power. Plot boundary (red line) corresponds to boundaries of domain $\Omega$ (cf. Figure 1).

**Table 7.** Dominant wind-direction case – evaluation of optimal layout. Relative power for 3 different wind directions comparing outputs of LES and Jensen Wake Model with $k_w = 0.036$.

| Wind direction (degrees) | Relative Power (LES) | Relative Power (Jensen Model) | Error |
|---|---|---|---|
| 0 | 93.08% | 91.91% | 1.17% |
| 7.5 | 94.27% | 93.21% | 1.06% |
| −7.5 | 93.44% | 92.43% | 1.02% |
| average $(0, 7.5, -7.5)$ | 93.67% | 92.51% | 1.08% |
| average $(-7.5 : 1.5 : 7.5)$ | – | 92.53% | – |

combines a global method with a gradient-based approach, as proposed by Réthoré et al. (2014), and exploring a large number of optimization starting points, may be required for studying this in more detail. This is an interesting topic for further research.

## 4 Conclusions

In the current work, we proposed a hybrid Jensen–LES approach for layout optimisation of wind-farms. The Jensen model is a
5   wake model that is sufficiently fast to allow in principle wind-farm optimisation over different wind-directions, and using global optimisation approaches that take into account the non-convex nature of the optimisation problem. Large-eddy simulations are much more accurate then the Jensen model, but are by orders of magnitude too slow to be used for wind-farm layout optimisation. Therefore, we introduce a nested optimisation approach in which the Jensen model is used as a surrogate model. In the inner loop, the Jensen model is used to perform the layout optimisation, while in an outer loop, the wake expansion
10   coefficient in the Jensen model is adapted to better fit LES results of the gradually evolving optimal layouts.

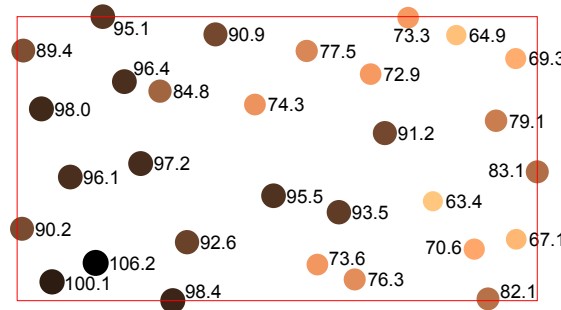

**Figure 9.** Optimal layout and relative power for the uniform wind-direction case (angle of $\pm 180°$). Relative power results are obtained from large-eddy simulations. Turbine locations are marked with colored disk: size and color are scaled with relative power. Plot boundary (red line) corresponds to boundaries of domain $\Omega$ (cf. Figure 1).

**Table 8.** Uniform $360°$ case – evaluation of optimal layout. Relative power for 8 different wind directions comparing outputs of LES and Jensen Wake Model with $k_w = 0.036$.

| Wind direction (degrees) | Relative Power (LES) | Relative Power (Jensen Model) | Error |
|:---:|:---:|:---:|:---:|
| 0 | 87.11% | 89.16% | −2.05% |
| 45 | 96.08% | 96.06% | 0.03% |
| 90 | 94.75% | 94.37% | 0.39% |
| 135 | 96.27% | 95.27% | 1.00% |
| 180 | 87.97% | 89.41% | −1.45% |
| −135 | 97.36% | 96.35% | 1.01% |
| −90 | 94.59% | 94.37% | 0.23% |
| −45 | 95.46% | 94.91% | 0.55% |
| average | 93.45% | 93.55% | −0.10% |

In the current study, layout optimisation of a wind-farm of 30 turbines on a $4\text{km} \times 3\text{km}$ area is considered. For this set-up, we found that an iterative fitting of the average wake expansion coefficient in the Jensen model during optimization to be sufficient, leading to errors below 1% for the optimal layout. For larger wind-farm layouts, wind-farm areas that are more complex, or including different atmospheric stratification regimes, it may be necessary to consider a more complex parametrization of the wake expansion coefficient. This may include dependence of the wake expansion coefficient on wind direction, Obhukov scale, or downstream location in the wind-farm. These are topics for further research.

Finally, the layouts found for the current set-up differed greatly depending on the wind-direction scenario. In case of a dominant wind directions, turbines were clustered together at the front and back of the wind-farm area, allowing for maximum wake recovery in between. For a $360°$ uniformly distributed wind rose, turbines are evenly spread out over the domain. However, this is a result from optimisation of energy yield only, given a number of turbines and wind-farm area, and the effect of wake–wake and wake–boundary-layer interaction. In practise, wind-farm layout optimisation is a multidisciplinary problem that includes effects and costs of turbine loading, costs of installation, maintenance, cabling, etc. The full inclusion of a hybrid Jensen–LES model in such an optimisation framework is also an important topic of further research.

*Acknowledgements.* This work is supported by the European Research Council (ActiveWindFarms, ERC grant No 306471), and the Flemish Science Foundation (Grant No. G.0376.1). Pieter Bauweraerts acknowledges support from Agency for Innovation by Science and Technology in Flanders (IWT). The computational resources and services used in this work were provided by the VSC (Flemish Supercomputer Center), funded by the Research Foundation – Flanders (FWO) and the Flemish Government – department EWI.

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
