# Peer review of "Wind-farm layout optimisation using a hybrid Jensen–LES approach"

_Wind Energy Science, 2016_

## Referee Comment (RC1) · Anonymous Referee #1 · 2 Jul 2016

General Comments

The paper "Wind-farm layout optimisation using a hybrid Jensen–LES approach" pursues an interesting hybrid approach to wind farm layout optimization that tunes the Jensen linear flow model to better agree with time averaged-LES results during optimization iterations. The layout optimization process uses the cross-entropy (CE) method that is fairly new to the wind farm community. The approach is novel, however I would like to see further discussion of the advantages/disadvantages of a sampling based approach like CE and a better characterization of the local/global nature of the solutions and convergence rates. The LES results also appear to need significantly more computation time as there is great variability in the turbine power production and unphysical structures are still visible in the LES time averaged velocity fields.

[Figure]

Specific Comments

Jensen model formulation:

1. The velocity deficit is introduced as \Delta U in equation 3 and then at the top of page 4 the turbine velocity is defined as U_{\infty} - \Delta U. I believe this should actually be U_{\infty} * (1 - \Delta U).

CE questions:

3. Is CE a global optimization method or local? Are there convergence rates or other ways to characterize its performance?

4. Does applying the site constraint by projecting the turbine onto the boundary change the underlying distribution you sample from? Does this affect convergence?

Jensen optimization:

5. Why use CE to fit k_w? With just a single parameter this seems like overkill.

6. There have been a number of recent developments in gradient based optimization of Jensen/FLORIS models (see papers by Gebraad, Ning, Fleming, etc). Can you comment on a sampling approach of CE vs gradient-based optimization methods?

7. Is the expansion coefficient optimization done to maximize agreement in power production on a per-turbine basis or simply total power output? Why not try to maximize agreement in the velocity field itself?

Results:

8. In Figure 3 the almost 15% range in relative power outputs for the first row of turbines is substantial and as noted by the authors requires a much longer averaging period. Tuning the Jensen model to match a 15% variability in turbine to turbine power could lead to incorrect values of k_w. While the authors argue that meteorological conditions would change before achieving a sufficient time window for the averages to
converge to 100%, this seems irrelevant for the purposes of fitting k_w since the goal is simply to produce the best time-averaged flow fields. If a longer LES time averaging period improves the power prediction, it should also change the optimal k_w values (presumably for the better).

9. In Figure 4 Jensen model results appear to have a velocity of almost 0 directly downwind of the first turbines. With $C\_T = 8/9$ as reported on page 6, $\Delta U$ can be at most 2/3. Using the corrected form for the velocity, this results in values of $u/u\_{\infty}$ that should not drop below 1/3. Have the authors implemented other modifications to the Jensen model? Or perhaps there is an error in the colormap?

10. In the lower left panel of Figure 4 there appear to be changes in the far wake that are noticeable in a number of turbines, for example in the wake of the two most upstream turbines. Are these numerical artifacts or is this a reflection of the wake off the ground? Further discussion would be helpful.

Optimization Results:

11. Are Figures 6, 7, 8, and 9 showing power results from the time averaged LES or Jensen model? The most upwind turbines are not producing 100% power, so perhaps it is LES?

12. Why are the optimization results for a single wind direction not symmetric about the midplane? Is this the global optima?

13. How sensitive are the optimization results to the choice of initial distributions or the samples chosen from a given distribution? Do you arrive at the same optima if you repeat the optimization process?

Technical Corrections

1. $\Omega$ in Eq. 2 is not defined.

2. Don't capitalize atmospheric boundary layer on line 24 page 3

3. Line 11 page 5 'trust coefficient' should be 'thrust coefficient'

4. Kilometer should be abbreviated 'km' not 'Km'

5. Second line in Table 1 should possibly read '(without fringe region)' instead of repeating '(with fringe region)'?

6. Would be convenient to list hub height velocity in Table 1.

7. Are the reported LES resolutions before or after the dealiasing is applied?

8. Line 5 page 11 describes Figure 3 as showing staggered, gradually staggered and 2 random layouts, but appears that the figure actually shows aligned, staggered, and 2 random layouts.

9. Page 19 line 5 'a iterative' should be 'an iterative'.

---

## Referee Comment (RC2) · Anonymous Referee #2 · 9 Jul 2016

This paper presents a new approach for the optimal micro-sitting of wind turbines problem. The main contribution of this work lays on the calibration (by running Large-Eddy Simulations) of the decay constant, Kw, used in Jensen's model to assess the wake effect. This approach is interesting but in my opinion it does not allow to overcome the main limitations of the Jensen's model, especially when assessing the multiple wake effect in large arrays.

The decay constant is usually determined by empirical observations. Therefore, running simulations to recalculate this constant doesn't seem to be a significant improvement on the wake model.

The problem description section needs to be improved. Also, it would be helpful to include a flowchart of the proposed methods.

When defining the optimisation problem in (2), the averaged Pi seems to be the same as the averaged Pi defined in (1), which according to the statement "T is a time averaging window that is sufficiently long to average out the turbulence effects." corresponds to a short-term average. However, when dealing with a planning problem the objective should be either maximise energy or long-term averaged power.
* * *

---

## Author Comment (AC1) · 4 Aug 2016

We thank the reviewer for his/her time, and the comments that were provided on our work. These are further discussed point by point, indicating in what way we intend to address them in a revised version of the manuscript (to be approved first by the editor).

1. **Reviewer:** *This paper presents a new approach for the optimal micro-sitting of wind turbines problem. The main contribution of this work lays on the calibration (by running Large-Eddy Simulations) of the decay constant, $K_w$, used in Jensen's model to assess the wake effect. This approach is interesting but in my opinion it does not allow to overcome the main limitations of the Jensen's model, especially when assessing the multiple wake effect in large arrays. The decay constant is*

*usually determined by empirical observations. Therefore, running simulations to recalculate this constant doesn't seem to be a significant improvement on the wake model.*

**Response:** We thank the reviewer for this comment. We realize that we may not have sufficiently explained our main objectives. As indicated by the reviewer, the Jensen model with a constant $k_w$ may not work well, e.g., in case of large arrays. However, this can be remedied by using a wake expansion coefficient that is not constant, but depends on streamwise distance into the farm, or on local turbulence intensity. An example is found in the work of Stevens, Gayme, & Meneveau (J. Renewable and Sustainable Energy 7, 023115, 2015), in which a top-down model is used to tune a streamwise dependent expansion coefficient, providing good results for extended farms with various alignment patterns. More generally, such a relation for $k_w$ may also depend on wind direction, and certainly atmospheric stability.

However, the problem then is how to fit $k_w$ a priori, in particular since the parameter will most certainly also depend on the wind-farm layout, which is not known before optimization. To this end, LES can be used in the approach proposed in our work. The main aim of our paper, is in showing how to do this, i.e. the main challenge is in making this computationally feasible, given the very high costs of performing LES. Our demonstration case itself is a moderate farm (in size), i.e. 30 turbines. In our particular case it turns out that is suffices to use a constant $k_w$ (something we didn't expect when we started our study), but our method does not preclude more complicated parametrizations for $k_w$. Indeed this may be relevant for larger farms, or farm areas that are much more irregular in shape.

We propose to better discuss this in our revised manuscript, in particular we will add one or two paragraphs making these points following the one but last paragraph of the current introduction.
2. **Reviewer:** *The problem description section needs to be improved. Also, it would be helpful to include a flowchart of the proposed methods.*

   **Response:** This is a good suggestion – we will add a flowchart with the different models and how they are linked, as suggested by the reviewer. See also next point for further discussion on problem description

3. **Reviewer:** *When defining the optimisation problem in (2), the averaged Pi seems to be the same as the averaged Pi defined in (1), which according to the statement "T is a time averaging window that is sufficiently long to average out the turbulence effects." corresponds to a short-term average. However, when dealing with a planning problem the objective should be either maximise energy or long-term averaged power.*

   **Response:** We agree with the reviewer. The power needs to be optimized for a much longer time horizon, i.e. over a period of years, and covering multiple wind directions and atmospheric conditions. Thus equation (2) essentially needs to be updated to include a summation (or integral) over all atmospheric conditions occurring (wind directions, stability classes, etc.). In the paper, we then basically first elaborate a case for one condition only, and later for a wind rose (but given one stability class only). In a revised version of the manuscript, we propose to address these issues both in the problem description section, as well as in the discussion and conclusion afterwards.

We hope that the approach suggested above for improving our manuscript is acceptable, so that we are allowed to proceed with a revision.

---

## Author Comment (AC2) · 5 Aug 2016

We thank the reviewer for the comprehensive comments that he/she provided on our work. These are further discussed point by point below, indicating in what way we intend to address them in a revised version of the manuscript (to be approved first by the editor) – we believe that this will be instrumental in further improving the manuscript.

**General comments**
**Reviewer:** *The paper "Wind-farm layout optimisation using a hybrid Jensen–LES approach" pursues an interesting hybrid approach to wind farm layout optimization that tunes the Jensen linear flow model to better agree with time averaged-LES results*

[Figure]

*during optimization iterations. The layout optimization process uses the cross-entropy (CE) method that is fairly new to the wind farm community.*

A. *The approach is novel, however I would like to see further discussion of the advantages/disadvantages of a sampling based approach like CE and a better characterization of the local/global nature of the solutions and convergence rates.*

**Response:** This is a good suggestion. See the discussion below under 'specific comments', answers to points 3–6 on how we propose to address this.

B. *The LES results also appear to need significantly more computation time as there is great variability in the turbine power production and unphysical structures are still visible in the LES time averaged velocity fields.*

**Response:** It is true that the LES flow field is not fully converged. We will include in the revision a detailed convergence analysis, integrating the flow field for a number of cases over a much longer time horizon. However, turbulent statistics converge slowly, in particular in neutral or unstable boundary layers where large turbulent structures with sizes of several kilometer are known to exist. Related time scales can be in the order of 100 to 500 seconds. The average flow field only converges with errors below 1% once a sufficient amount of these time scales are covered, requiring typically 10 to 20 hours of simulation time, also depending on the level of turbulence intensity. This may make an LES-based optimization approach (even using our hybrid approach) unfeasible. Therefore, we keep our simulation time shorter, and only focus on convergence of the total farm power, since averaging over the number turbines in itself already reduces the variance and more so for a larger farm (roughly proportional to $N_t^{-1/2}$, also depending on the turbine's relative positions). Thus, in particular for large optimization problems (many turbines), the LES cost can be reduced considerably. Moreover, when

considering optimization given a wind rose and changing atmospheric conditions over the year, this type of variability will be even reduced more.

In the revision, we propose to discuss these issues in more detail in the LES methodology section, including a detailed convergence analysis.

**Specific comments**

1. **Reviewer:** *The velocity deficit is introduced as $\Delta U$ in equation 3 and then at the top of page 4 the turbine velocity is defined as $U_\infty - \Delta U$. I believe this should actually be $U_\infty * (1 - \Delta U)$.*

   **Response:** Thanks for pointing this out. This is indeed correct given the definition of $\Delta U$ in equation 3. We'll correct this by multiplying Eq. (3) with $U_\infty$ – in this case the turbine velocity correctly corresponds to $U_\infty - \Delta U$. [we prefer correcting it this way, so that $\Delta U$ has the dimension of a velocity as suggested by the use of the symbol $U$]

2. There was no point 2 in the reviewer's report [we just insert it here to keep the numbering the same]

3. **Reviewer:** *Is CE a global optimization method or local? Are there convergence rates or other ways to characterize its performance?*

   **Response:** The CE method is a global optimization method. First of all, we recall that global optimization is a Np hard problem, so that algebraic convergence rates such as established for convex problems cannot be determined. The

method formally converges to the global optimum for $\alpha \to 0$ (see, e.g., Costa et al. Operations Research Letters 2007). In practise, the selection of $\alpha$ is a trade-off between convergence speed of the algorithm and formal convergence to the global optimum. Similarly, also the selection of the total number of iterations corresponds to such a trade-off.

In the manuscript, we propose to better discuss these points, and in particular point out that the parameters employed in the method require some tuning for good convergence.

4. **Reviewer:** *Does applying the site constraint by projecting the turbine onto the boundary change the underlying distribution you sample from? Does this affect convergence?*

**Response:** It will indeed change the underlying distribution – our guess is that this does not hurt the method, as we could also have used the approach for the nonlinear constraint (keeping the original distribution). In practice, we found that use of the projection for the boundary constraints was beneficial for the convergence speed of our optimization. We propose to add a cautionary note and discussion on this in the manuscript when introducing the CE method.

5. **Reviewer:** *Why use CE to fit $k_w$? With just a single parameter this seems like overkill.*

**Response:** There is no particular reason – we just stuck to the same method that we were using for the layout optimization. In fact, any type of optimization method may be feasible – the main intention of our work is not related to the optimization method itself, but rather on the hybrid combination of LES and Jensen as a surrogate model. We will add a paragraph in the methodology section indicating that the approach can be combined with any type of optimization method

that is deemed fit by the user.

6. **Reviewer:** *There have been a number of recent developments in gradient based optimization of Jensen/FLORIS models (see papers by Gebraad, Ning, Fleming, etc). Can you comment on a sampling approach of CE vs gradient-based optimization methods?*

**Response:** Thanks for pointing this out. Indeed, it is also possible to use a gradient-based approach for the optimization, and we will refer to the papers mentioned. The advantage of a gradient-based approach is that it is very fast compared to global methods (usually with linear or superlinear convergence rates); their disadvantage is that they only converge to the nearest local optimum. Given the fact that Jensen evaluations are extremely cheap, we believe that using a global method is feasible and potentially beneficial. However, it could be nice to use a global method in a first prediction step, and subsequently use the result of this as a starting point for a gradient based method, since a global method is usually not stopped based on a convergence criterion, but rather based on a maximum number of iterations.

Following the reviewers comment, we did experiment with a gradient-based approach using the Matlab `fmincon` function starting from the optimum of the global method. Unfortunately, including all the nonlinear distance constraints turned out to not work properly (there are 435 of them for 30 turbines). However, leaving them out (thus allowing turbines to be placed closer together) did not alter the overall power predicted by the Jensen model significantly (i.e. power increased by 0.4% only), leading us to the conclusion that our CE optimum is very close to a (local) optimum.

7. **Reviewer:** *Is the expansion coefficient optimization done to maximize agreement in power production on a per-turbine basis or simply total power output? Why not*

*try to maximize agreement in the velocity field itself?*

**Response:** The coefficient is optimized based on total power production. This is OK, since we optimize not just for one case, but for a series of about at least 10 different (partly random generated) cases. The reason why we do not maximize agreement on a per turbine basis is related to the convergence speed, and the need to keep LES costs under control (cf. discussion above under A.). We will better discuss this in the revision.

8. **Reviewer:** *In Figure 3 the almost 15% range in relative power outputs for the first row of turbines is substantial and as noted by the authors requires a much longer averaging period. Tuning the Jensen model to match a 15% variability in turbine to turbine power could lead to incorrect values of $k_w$. While the authors argue that meteorological conditions would change before achieving a sufficient time window for the averages to converge to 100%, this seems irrelevant for the purposes of fitting $k_w$ since the goal is simply to produce the best time-averaged flow fields. If a longer LES time averaging period improves the power prediction, it should also change the optimal $k_w$ values (presumably for the better).*

**Response:** As indicated under point A. above, we will extend our LES convergence analysis. In addition, we would like to point out that $k_w$ is fitted using the mean power, but over 10 to 20 different layouts. Thus the variability of 15% at turbine level mentioned by the reviewer is largely averaged out (over 30 turbines times at least 10 layouts). Moreover, we further found that our optimization results are rather robust against changes in $k_w$ – in fact optimization results do not significantly alter when changing the second digit in the expansion coefficient. We will also include this element in the discussion.

9. **Reviewer:** *In Figure 4 Jensen model results appear to have a velocity of almost 0*

*directly downwind of the first turbines. With $C_T = 8/9$ as reported on page 6, $\Delta U$ can be at most 2/3. Using the corrected form for the velocity, this results in values of $u/u_\infty$ that should not drop below 1/3. Have the authors implemented other modifications to the Jensen model? Or perhaps there is an error in the colormap?*

**Response:** Thanks for noticing this – there was a postprocessing error in the figure (in fact, we erroneously plotted $(U/U_\infty)^3$). Attached is the corrected figure.

10. **Reviewer:** *In the lower left panel of Figure 4 there appear to be changes in the far wake that are noticeable in a number of turbines, for example in the wake of the two most upstream turbines. Are these numerical artifacts or is this a reflection of the wake off the ground? Further discussion would be helpful.*

**Response:** This is indeed related to wake reflection – we use mirror turbines to mimic this, as usually done in the Jensen model. We will further briefly discuss this when discussing Figure 4 in section 3.2

11. **Reviewer:** *Are Figures 6, 7, 8, and 9 showing power results from the time averaged LES or Jensen model? The most upwind turbines are not producing 100% power, so perhaps it is LES?*

**Response:** These figures are showing power results from time-averaged LES. The differences between turbines, and upwind turbines not all showing 100% power is related to the convergence of the LES. As discussed above under $A$, we will include a more detailed convergence analysis of our results, and also comment on this in our result discussion.

12. **Reviewer:** *Why are the optimization results for a single wind direction not symmetric about the midplane? Is this the global optima?*

**Response:** As far as we can see, there is no reason why they should. There may be cases, also depending on the number of turbines, were the global optimum is symmetric, but this does not have to be. Using gradient-based optimization (starting from the CE optimum - cf. point 6), the turbines positions shifted a bit, but we also did not find a symmetric optimum.

13. **Reviewer:** *How sensitive are the optimization results to the choice of initial distributions or the samples chosen from a given distribution? Do you arrive at the same optima if you repeat the optimization process?*

    **Response:** We found that turbine positions do slightly shift depending on the choice of initial distribution, but this does not significantly influence the value of the power extraction. In fact, we found for the cases that we explored that there is a very flat 'valley' around the optimal arrangement pattern. This indicates that in a real ecomonic-driven wind-farm optimization other parameters, such as cost of cabling, foundations (given soil conditions), etc. will play a significant role in the final layout. We will extend the discussion at the end of section 3.3 with these elements.

**Technical corrections**

1 –5,8,9: Thanks for these corrections. We'll apply them to the manuscript.

6 **Reviewer:** *Would be convenient to list hub height velocity in Table 1.* **Response:** the hub-height velocity is not really relevant, since we can rescale our simulations using the friction velocity - simulations remain dynamically equivalent as long as

$z_0/z_h$ remains the same, time is scaled with $z_h/u_*$, and $C'_T$ remains constant. Our particular LES simulations yield $u_h/u_* = 17.5$, so a hubheight velocity of e.g. 10 $m/s$ would correspond to $u_* = 0.57$. We will better explain this in the paper, and also in particular explain better the averaging times in nondimensional units, since in fact they are currently expressed in units of $z_h/100/u_*$.

7 **Reviewer:** *Are the reported LES resolutions before or after the dealiasing is applied?*
**Response:** After – we apply the $3/2$ rule. Our grid $256 \times 256 \times 80$ is extended to a $384 \times 384 \times 80$ in real space to perform all nonlinear operations, and brought back to $256 \times 256 \times 80$ in Fourier space afterwards. We'll clarify in section 2.3.

We hope that the approach suggested above for improving our manuscript is acceptable, so that we are allowed to proceed with a revision.
* * *
[Figure]

[Figure]

**Fig. 1.**

---

## Author Response (AR2)

**Changes in response to the final corrections requested by the Editor-in-Chief**

**Editor-in-Chief:** *Thanks for the revision. I think the paper still misses some essential discussion. In most of the optimizations, even those with an even directional wind distribution, the turbines do not end up in the corners of the domain. This is to be expected, especially because cable length is not minimized. If you look at the paper by Rethore et al (2013) Wind Energy 10.1002/we.1667 you see a clear tendency of the turbines to go for the corners. Why do you not observe that and compare to their work? Wouldn't you expect to see turbines in the corners or have I (and Rethore) misunderstood something? Has it to do with a discretization of the angular distribution?*

**Response:** This is an interesting point. Looking at the work of Rethore et al (2013), there is indeed a case, i.e. the Stags Holt / Coldham case (in which line costs appear not to dominate) and with turbines ending up at all corners of the domain. However, this seems not to be consistently present in all reported optimization results in literature. Some papers show similar results (see, e.g. Feng & Shen 2015), while others do not always end up with turbines at the corners (e.g., Du Pont and Cagan, 2012) – all considering a number of cases with a uniformly 360°wind rose. Currently, we don't know whether this is possibly dependent on number of turbines and size and shape of the wind-farm domain, or whether turbines should always end up at the corners (in view of the complexity of the problem, we're not sure that this can be just concluded on intuition only). At the same time, it may also be possible that turbines not ending at the corners depends on convergence of the optimization method, the existence of many local optimums, so that in the global optimum, turbines are indeed always at the corners – given the non-convex nature of the problem, establishing this rigourously may not even be possible, and we did not find any studies in which this issue is explicitly resolved. In any case, we agree that this warrants some discussion in our manuscript. On page 20, last paragraph of §3.4, we extended the discussion as follows:

"The optimal layout itself is shown in Figure 9. In contrast to the layout found for the dominant wind direction, now turbines are spread out much more evenly throughout the domain. Moreover, a number of turbines, i.e. seven, are located on the domain boundary. We remark here, that for similar optimization cases in literature, turbines sometimes end up at the domain corners (see, e.g., Rethore et al 2013, or Feng & Shen 2015), but this is not the case for all studies (e.g., Du Pont & Cagan 2012). Currently, we are not sure whether this is possibly related to domain shape, size, and number of turbines, or whether this is related to the existence of local optimums, or convergence of the optimization method. Using a hybrid method that combines a global method with a gradient-based approach, as proposed by Rethore et al 2013, and exploring a large number of optimization starting points, may be required for studying this in more detail. This is an interesting topic for further research."

Finally, we reiterate that the scope of our current manuscript is on combining LES with the Jensen model for layout optimization with focus on the formulation of an approach that is computationally feasible (given the large costs of LES). It is not our intention to make an extensive comparison of optimization methods, and any kind of method can be straightforwardly used in the approach instead of the CE method (this is also explicitly discussed in the paper). Providing strong results on the precise properties of optimal layouts is a study in itself. Given the non-convex nature of the problem, it may be interesting to, e.g., look at the optimums of convex subproblems that arise when solving the optimization with an SQP approach, compare various global+gradient-based combinations for a large number of starting points, etc., but this is out of the scope of our current work. Therefore, given the additional discussion provided above, we hope that the manuscript can now be accepted in its current form.

―――――――――――――――

[1] Du Pont, B. L. and Cagan, J.: An extended pattern search approach to wind farm layout optimization, Journal of Mechanical Design, 134, 081002, 2012.

[2] Feng, J. and Shen, W. Z.: Solving the wind farm layout optimization problem using random search algorithm, Renewable Energy, 78, 182–192, 2015.

[3] Réthoré, P.-E., Fuglsang, P., Larsen, G. C., Buhl, T., Larsen, T. J., and Madsen, H. A.: TOPFARM: Multi-fidelity optimization of wind farms, Wind Energy, 17, 1797–1816, 2014.